# Description of global innovative methods in developing the WHO Community Engagement Package

Yolanda Vargas Bayugo,[1] Meredith Labarda,[2] Jose Rene Bagani Cruz [ID] ,[3] Jana Deborah Mier-Alpaño [ID] ,[3] Pauline Marie Padilla Tiangco,[3] Ukam Ebe Oyene [ID] ,[1] Semeeh Akinwale Omoleke,[1,4] Allan Ulitin [ID] ,[5] Alberto Ong Jr,[6] Marvinson See Fajardo [ID] ,[3] Maria Isabel Echavarria [ID] ,[7,8] Jackeline Alger,[9,10] Don Mathanga,[11] Barwani Khaura Msiska,[12] Obinna Ikechukwu Ekwunife,[13,14] Obioma Nwaorgu,[14,15] Lorena Abella Lizcano,[7,8] Natalia Gomez Quenguan [ID] ,[7,8] Claudia Ivette Nieto Anderson [ID] ,[16] Briana Yasmin Beltran,[17] Elsy Denia Carcamo Rodriguez,[18] Eduardo Salomón Núñez [ID] ,[19,20] Vera Nkosi-Kholimeliwa,[21] Glory Mwafulirwa-Kabaghe,[22] Noel Juban[23]

NJ since deceased.

For numbered affiliations see end of article.

**Correspondence to**
Dr Meredith Labarda;
mdlabarda@up.edu.ph

## ABSTRACT

**Objectives** Development of a Community Engagement Package composed of (1) database of community engagement (CE) experiences from different contexts, (2) CE learning package of lessons and tools presented as online modules, and (3) CE workshop package for identifying CE experiences to enrich the CE database and ensure regular update of learning resources. The package aims to guide practitioners to promote local action and enhance skills for CE.

**Setting and participants** The packages were co-created with diverse teams from WHO, Social Innovation in Health Initiative, UNICEF, community practitioners, and other partners providing synergistic contributions and bridging existing silos.

**Methods** The design process of the package was anchored on CE principles. Literature search was performed using standardised search terms through global and regional databases. Interviews with CE practitioners were also conducted.

**Results** A total of 356 cases were found to fit the inclusion criteria and proceeded to data extraction and thematic analysis. Themes were organised according to rationale, key points and insights, facilitators of CE and barriers to CE. Principles and standards of CE in various contexts served as a foundation for the CE learning package. The package comprises four modules organised by major themes such as mobilising communities, strengthening health systems, CE in health emergencies and CE as a driver for health equity.

**Conclusion** After pilot implementation, tools and resources were made available for training and continuous collection of novel CE lessons and experiences from diverse socio-geographical contexts.

## INTRODUCTION

There is an increasing necessity to redouble efforts using innovative approaches to bolster community engagement (CE) in the global health setting. Emergencies, including the COVID-19 pandemic, severely disrupted prevention and treatment services for non-communicable diseases, malaria and other interventions.[1–4] This has compounded health inequities and widened the gap across populations. The complexities brought about by these health problems make community participation in co-creating innovative solutions to these challenges even more critical. The shift to people-centred approaches, as highlighted in the revised WHO risk communication and community

engagement (RCCE) strategy,[5 6] is imperative as CE can make a considerable difference in health outcomes and capacitate communities to deal with health challenges and their determinants.[7–9] The response to the Nepal earthquake and similar experiences made clear that people-centred design and leadership in addressing problems facilitate more efficient use of resources, strengthen coordination and build local capacities.[10] The WHO, UNICEF and development partners support CE with resource mobilisation, information, and training with various outcomes and competencies.[11] However, there is no harmonised CE documentation package based on local contexts for training. This project was initiated to guide health practitioners in promoting local action, and to facilitate involvement, training, and synergies across health and development sectors to achieve collective outputs and outcomes.[12–15] It responds to the need to invest in effective social innovations grounded on CE, which use bottom-up approaches and draw on strengths of individuals, communities and institutions while promoting synergies across sectors.[16–18]

⇒ This project fills a need for a harmonised CE documentation package for training based on different local contexts and with a broad range of health and social development activities including health emergencies, routine immunisation, neglected tropical diseases, city and urban development, nutritional interventions and disaster risk management.

⇒ The CE cases identified were limited to those in English, French and Spanish. Future researches can explore relevant documented and undocumented experiences in other languages.

⇒ The CEP was developed and tested primarily through online environments and might need adjustment for in-person implementation.

## The WHO CE package

The WHO Department of Country Readiness Strengthening conceptualised and initiated the Community Engagement Package (CEP) project based on consultations within WHO Regional Offices and Headquarters. The CEP project[19] developed a database of CE experiences, a CE learning package (CELP) and a CE workshop package (CEWP) based on a broad scope of CE experiences in different settings. The compiled cases can guide programme managers, CE practitioners, in-service medical and non-medical trainees, non-governmental organisation staff and multidisciplinary teams to sharpen their skills in the CE approach.

## CEP project design and components

The design of the CEP involved the creation of a database of relevant CE cases. These cases were categorised and analysed, and themes and concepts were used to develop the CELP with contributions from CE subject matter experts (SMEs). The CEWP was designed to document 'newer' CE experiences that can be incorporated into the database, ensuring regular updates of the learning resources (see figure 1). Table 1 summarises the three components of the CEP.

Given the uniqueness, relevance and value of the harmonised CEP in the context of health emergencies and the overall global health sphere, this paper seeks to document the processes and the innovative ways by which the CEP was developed at the height of COVID-19 restrictions.

## METHODS
### Patient and public involvement

The conceptualisation, design, and conduct of the CEP involved participation and co-creation among colleagues and potential end users in the WHO, Social Innovation in Health Initiative (SIHI) hubs, UNICEF and other implementing partners, and community practitioners and frontline responders.

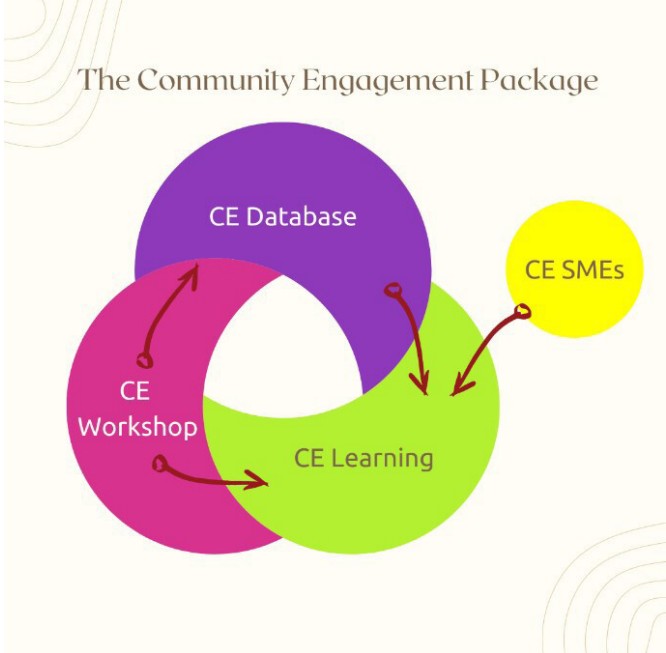

**Figure 1** WHO Community Engagement (CE) Package components and relationships. SMEs, subject matter experts.

## CEP human resource infrastructure and way of working

The overall project methodology was anchored on CE principles and processes. Colleagues in WHO (headquarters and regions) participated in the CEP project. The SIHI global network contributed substantially to the realisation of the CEP.

### WHO CEP working group

The design of the CEP project came about after consultations with WHO colleagues involved in CE work, bringing

**Table 1** Descriptions of the components of the WHO Community Engagement Package

| | |
|---|---|
| Community engagement database | Organised collection of data and documentation of community engagement experiences, practices, and approaches in different regions and contexts. |
| Community engagement learning package | Curation of community engagement lessons and tools presented as online (asynchronous) modules designed to capacitate learners on basic concepts, principles, and applications of community engagement, and explore best practice experiences in solving health problems and promoting health through community engagement. |
| Community engagement workshop package | Provides tools and templates for identifying community engagement experiences in a workshop format. The contents are similar to the community engagement learning package, with a special focus on documenting 'new' community engagement experiences and their nuances, and a walk-through of using and submitting case studies for the community engagement database. |

in experiences of WHO working with communities in different contexts and settings.[19] These colleagues work in different thematic areas: health promotion, social determinants of health, health systems, disaster risk reduction, risk communication, healthy cities, community readiness and resilience, and population-based focused work. As the CEP design was drafted, a working group (WG) was established to provide technical advice and CE resources related to their respective areas of work. Regular WG meetings were conducted to ensure that they had updated information and an opportunity to provide feedback to improve the package. Some members of the WG also participated as resource persons in the CELP.

The WG also consulted and regularly updated the RCCE Collective Services, which is composed of WHO, UNICEF, International Federation of Red Cross and Red Crescent Societies, and Global Outbreak Alert and Response Network. UNICEF provided inputs regarding training.

### SIHI global network

The SIHI Philippines hub is the main implementing agency of the project. It is part of the SIHI global network of research hubs and other partners supported by TDR, the Special Programme for Research and Training in Tropical Diseases. SIHI hubs have expertise and experience documenting social innovations from communities and communicating these innovations with stakeholders.

Led by the SIHI Philippines, the SIHI hubs based in Colombia, Honduras, Malawi, Nigeria and South Africa also participated in this project. Together, they gathered published and grey literature on CE and were involved in the development of the search terms and selection criteria, case abstracts and identification of themes. SIHI Philippines spearheaded the development of the prototype learning and workshop packages and facilitated regular virtual meetings with the other hubs and WHO staff for updates and consultation.

### Development of the components of the CEP

The development of the components of the CEP can be characterised as iterative, collaborative and comprehensive and can be considered 'community engagement in practice'.

### Development of the CE database

The CE database is an organised collection of data and documentation of CE practices, experiences, and approaches used in different regions and contexts. Systematic search was done to gather and organise these, integrating multistakeholder and consultative approaches across the SIHI global network and key partners from WHO.

### *Search for materials on CE*

This phase identified materials that document experiences about CE in programmes that address health or the social determinants of health. The search procedures were developed and co-created with SIHI hubs and the WHO using the 'system lens' principles and a bottom-up approach. Methods were refined as feedback was collected during implementation.

A standard procedure was prescribed for literature search to ensure the quality of cases found and maximise use of search platforms. For published literature (ie, case reports/series, review articles, research papers, journal articles), searches in PubMed, Google Scholar, Hinari, Research Gate, Scopus, Embase and LILACS were conducted. Other significant local and regional repositories were also explored.

The following standard search terms were used:

These terms were also translated to French and Spanish and additional terms for a geographical location were also added to focus searches in these areas.

For grey literature (ie, newsletters, unpublished reports or limited distribution, theses, conference papers/presentations, books and others), general search engines were used and academic and professional networks were tapped. Materials in languages other than English were included, with interpretation assistance from the SIHI network. Audiovisual materials were collected from credible organisational partners of WHO and SIHI, sources recommended by these organisations, and verified social media accounts and websites.

Interviews, surveys and correspondence with CE practitioners were facilitated to identify undocumented CE practices. Academic and professional networks of the SIHI network, WHO and partners were engaged in identifying undocumented CE practices for inclusion. Virtual communication technologies were used because of travel restrictions. Recordings or transcripts were obtained for documentation. The reviews were conducted by the project staff and SIHI hubs in coordination with the WG.

Following Preferred Reporting Items for Systematic Reviews and Meta-Analyses' recommended process flow, materials collected were screened initially through the title and abstract, when available. These were then assessed based on the selection criteria.

### *Selection criteria*

A set of criteria (table 2) was developed to standardise relevant CE cases that were entered into the database. This was based on inputs from various stakeholders and was finalised with consensus from WHO and the participating SIHI hubs. Definitions of specific terms also provided additional guidance.

### *Writing case summaries*

A summary was written for each identified case including the project's name, implementing institution, number of years the project was implemented, implementation site and health issues/topic addressed. The rationale, objectives, intervention, outcomes, lessons, challenges, and factors promoting and/or impeding CE were abstracted. Social innovations, if any, were included.

**Table 2** Inclusion criteria and guiding definitions for the selection of community engagement materials

**Inclusion criteria**

Documented in reputable sources or can provide information/documentation for the assessment of validity

Articles published in the last 10 years or undocumented experiences active within the last 10 years

All community engagement criteria are met:

► Captures or documents experience on community engagement addressing a health need or social determinants of health
► Uses a participatory approach and active two-way communication using language appropriate for different actors and stakeholders
► Encourages collaboration/synergies and sharing of expertise with various stakeholders and sectors, mainly, but not limited to, marginalised groups to improve capacities
► Involves the community in the different phases of implementation of the intervention/strategy such as planning, context analysis, decision-making, research, monitoring, evaluation and/or learning to ensure inclusive representation, maximum participation and uncompromised consultation
► Builds and sustains trust within the community
► To simplify the assessment of trust, the following criteria, based on the work of Di Napoli et al,[20] have been adopted. At least two of the four criteria must be met to indicate trust with the community:
 – Presence of interest and competence in offering services that support the community's needs and allow the realisation of the community members' aspirations
 – Community members are willing to participate in the improvement of the community through their effort of contribution of valuable resources
 – Community members find pleasure and meaning in spending their time participating
 – Community members expect that the engagement will improve future resources related to security, decision-making, participation and achieving their goals

**Definitions of terms**

| | |
|---|---|
| Communities | Groups of people who may or may not be spatially connected, but share common interests, concerns or identities. These communities could be local, national or international, with specific or broad interests[21] |
| Community engagement | The process of working collaboratively with and through groups of people affiliated by geographical proximity, special interest or similar situations to address issues affecting the well-being of those people[22] The process of developing relationships that enable stakeholders to work together to address health-related issues and promote wellbeing to achieve positive health impact and outcomes[23] |
| Social determinants of health | Non-medical factors that influence health outcomes. They are circumstances where 'people are born, grow, work, live, and age, and the wider set of forces and systems shaping the conditions of daily life'[24] |
| Trust | Positive expectations of community members toward the current and future opportunities they perceive in their local community, namely the place where they live and interact[20] Building purposeful and compassionate relationships through a resilient and community-competent health workforce that adapts to the needs and preferences of the people they serve[25] |

## Compilation of materials

All selected and created documents were uploaded to the project's Google Drive and kept in storage, pending migration to a WHO repository for the database, CELP and CEWP.

## Analysis and identification of common themes

Content analysis of the summaries and other data extracted from the screened materials was done using open coding. Key ideas and nuances were identified and grouped into categories and themes. These were then used to tag and organise the materials in the database.

## Development of the CELP

The CELP is a curation of CE lessons and tools presented as online (asynchronous) modules designed to capacitate learners on basic concepts, principles and applications of CE, and explore best practice experiences in solving health problems and promoting health through CE. In-depth analysis done with the contents of the database identified important CE principles, practices, lessons, challenges, and barriers encountered in different contexts and regions. Existing CE frameworks, toolkits and guides

were also surveyed. Emerging themes and concepts were used as the basis for the development of the CELP. SMEs contributed to the contents of the CELP designed to be delivered in an online learning management system.

Initial outline and plans for the CELP were also vetted among the CEP WG, and stakeholders and partners who have extensive experience in engaging and mobilising communities, both at the regional and global levels. Comments, critiques, suggestions, and recommendations that emerged from the series of vetting processes further shaped and enhanced the content of the learning package.

## Development of the CEWP

The CEWP was developed as a complementary strategy to the CELP, highlighting important topics and practical activities that might be useful for participants to enhance their CE practice. It was initially designed for face-to-face engagements, but because of the restrictions brought about by the pandemic, the pilot implementation was done online. The package materials were made into a downloadable format that can be adapted in either online

or face-to-face settings. Different iterations of the activity design were developed based on the different possible country contexts, using the input from SIHI networks and frontline responders engaging specific issues and populations—migrants, indigenous populations, people living with disabilities, women, elderly and youths.

## Testing the learning and workshop packages

Prototypes of the packages were tested among stakeholders, particularly community mobilisers, public health practitioners and other potential end users.

An online platform was created to test the online learning package. Pilot participants were selected using criteria that facilitated the inclusion of different groups and were invited to undergo the online asynchronous training. Feedback from the participants was obtained through online evaluation forms and was used to guide the revision of the training design.

Pilot testing for the workshop package was conducted in two phases through an online video conferencing platform. The first phase was implemented among participants from the Philippines. The pilot run tested the regional applicability and impact of the materials and content. The second phase was conducted among a global set of participants, which tested its universal applicability and impact. In both phases, user experiences were collected and used to refine the packages.

## Limitations in conducting the CEP activities/process

All engagements and coordination for this project were done remotely using online platforms due to the restrictions brought about by the COVID-19 pandemic. The team ensured that participatory approaches were reinforced and the voices of CE practitioners were incorporated in the CEP.

# RESULTS
## CE database

A database of experiences on CE was developed across public health in different settings. WHO and partners identified relevant resources that captured CE experiences, using the prescribed inclusion criteria. Materials in various formats (documents, videos, etc) that highlighted the practices, lessons and challenges in working with the communities were compiled. The documents and related materials are in English, Spanish and French. Summaries of documented CE cases are available in English.

## Categories of cases in the CE database

There are 356 cases in the database (290 identified from published literature, 57 from grey literature and 9 from CE practitioner interviews) from all six WHO regions, categorised according to the health topic (table 3). In addition, a total of 56 cases dealing with health emergencies were identified with 30 cases on COVID-19, 12 on Ebola, 9 on environmental risk and disaster, and 5 on humanitarian crises.

## CE practitioner interviews

Seven CE practitioner interviews were conducted—five interviewees from African Region (AFRO), one each from Pan American Region (PAHO) and Western Pacific Region (WPRO). These interviews identified nine unpublished CE experiences and explored CE strategies and dynamics and how that influenced the sustainability of health interventions.

## Thematic analysis

The case summaries were coded and analysed, capturing themes from the rationale for CE, key insights, facilitating factors and barriers. The documentation of the thematic analysis is available in an additional document

**Table 3** Distribution of cases according to health topic and the WHO regions

| Health topic category | Number of cases per WHO region | | | | | | |
| | AFR | EMR | EUR | PAHO | SEAR | WPR | Total |
|---|---|---|---|---|---|---|---|
| Communicable diseases | 66 | 10 | 2 | 20 | 14 | 21 | 133 |
| Primary healthcare | 9 | 2 | 11 | 13 | 6 | 8 | 49 |
| Maternal & child health | 9 | 1 | 2 | 5 | 5 | 3 | 25 |
| WASH | 6 | 0 | 1 | 3 | 1 | 0 | 11 |
| Sexual & reproductive health | 3 | 2 | 2 | 4 | 1 | 2 | 14 |
| Social determinant of health | 1 | 5 | 13 | 27 | 7 | 3 | 56 |
| Mental health | 0 | 3 | 1 | 5 | 1 | 4 | 14 |
| NCDs | 1 | 3 | 4 | 3 | 8 | 11 | 30 |
| Nutrition | 0 | 0 | 0 | 2 | 2 | 2 | 6 |
| Others | 3 | 0 | 5 | 3 | 5 | 2 | 18 |
| Total | 98 | 26 | 41 | 85 | 50 | 56 | 356 |

AFR, African Region; EMR, Eastern Mediterranean Region; EUR, European Region; NCDs, non-communicable diseases; PAHO, Pan American Region; SEAR, South East Asian Region; WASH, Water, sanitation and hygiene; WPR, Western Pacific Region.

**Table 4** Summary of themes from the community engagement cases

| Rationale for community engagement | Contextual and health system challenges<br>Health and social goals<br>Mechanisms |
| --- | --- |
| Key points and insights | Community mobilisation<br>Individual and community agency<br>Multistakeholder engagement<br>Multidirectional communication<br>Building on local capacity<br>Access, acceptability and adaptation<br>Inclusion<br>Sustainability<br>Participatory research<br>Basic principles |
| Facilitators of community engagement | Adapting the intervention<br>Applying participatory principles and approaches<br>Maximising reach and access<br>Using support mechanisms |
| Barriers to community engagement | Societal and contextual issues<br>Challenges with leadership<br>Weak health system<br>Challenges in encouraging and sustaining participation<br>Inadequate reach and access<br>Knowledge/information gaps<br>Lack of trust<br>Issues in communication<br>Inadequate or improper allocation of resources<br>Organisational and logistic problems<br>Challenges on the sustainability and generalisability of the project<br>Timing and duration of community engagement |

in the database. Table 4 presents the thematic areas that emerged from the review of the cases.

### Community engagement learning package

From the CE materials collected, the CELP was developed and anchored on basic principles and standards of CE and grounded on actual experiences in working with communities in different contexts and settings. The CELP includes four self-instructional modules that participants may complete independently or as a ladder-type course. Each module presents basic frameworks and concepts of CE in relation to the theme of that module and are then tied to real-world examples of CE in different contexts (see table 5). Target learners include early to mid-level professionals and practitioners applying CE in their work who may come from various disciplines such as medical and health sciences, public health, public policy and administration, programme management, social development and other social sciences. Students both at the undergraduate and postgraduate levels of any higher education institution, from various disciplines as mentioned above, may also benefit from the modules.

The pilot participants found the CELP to be comprehensive in terms of content and with a user-friendly format. They appreciated how other concepts in public health were linked to CE. They suggested more practical applications and specific how-to's, and assessment activities with immediate feedback. These were all taken into consideration in the revision of the modules.

### Community engagement workshop package

The CEWP provides tools and templates for identifying other CE experiences in a workshop format. The contents are similar to the CELP, with a special focus on documenting 'new' CE experiences and a walk-through of using and submitting case studies for the CE database. The target participants are practitioners who are interested in sharing their CE experiences. The CEWP allows the continuous collection of evidence and discussions with stakeholders on CE principles, practices and frameworks. These resources will be catalogued, categorised, and used to update the database and the learning and workshop packages.

Participants and observers of the CEWP pilot were satisfied with the introduction and ice-breaking activities which set the stage for conducive training sessions. Participants also expressed satisfaction on the content, pointing out that the workshop addressed aspects of CE not previously considered. The topics of the training were noted to be far-reaching, covering several CE frameworks, with good video presentations. Participants were able to relate the lessons and case studies to their experiences. They pointed out a few areas of improvement, including the need for adequate time to study the cases prior to the synchronous online sessions and more breakout sessions for participants to raise issues and ensure more diverse voices and opinions. They also recommended that the frameworks need to further emphasise listening and understanding community perspectives right from the start of the engagement.

### DISCUSSION

The CEP and its development showcase innovative elements in the project design, the human resources involved and way of working, and the inter-relationships of the different CEP components.

The CEP conceptualisation and design involved broad consultations and co-creation with a community of diverse teams of WHO, SIHI hubs, UNICEF, and other implementing partners and frontline responders. The process and products of the package were vetted among stakeholders and partners at the regional and global levels. In addition, community practitioners were consulted regarding the screening criteria of cases to be included in the database, shared undocumented CE practices, and participated in the pilots of the learning and workshop packages to provide user feedback. This multistakeholder consultative processes allowed for the creation of a grounded, contextualised, relevant and integrated package.

Working on the CEP project during the COVID-19 pandemic did not deter the WHO and SIHI from

**Table 5** Modules of the community engagement learning package

| Module title | Main framework/s used | Sample cases used |
|---|---|---|
| Module 1: Engaging and Mobilizing Communities for Health and Development | WHO community engagement framework for quality, people-centred and resilient health services[23]<br>Community engagement: a health promotion guide for universal health coverage in the hands of the people[26] | Setting health priorities in a community: a case example<br>Sousa et al[27]<br>Participatory learning and action to address type 2 diabetes in rural Bangladesh: a qualitative process evaluation<br>Morrison et al[28]<br>Community engagement in outbreak response: lessons from the 2014–2016 Ebola outbreak in Sierra Leone<br>Bedson et al[29]<br>'What works here doesn't work there': The significance of local context for a sustainable and replicable asset-based community intervention aimed at promoting social interaction in later life<br>Wildman et al [30] |
| Module 2: Strengthening Health Systems through Community Engagement | Systems thinking for health systems strengthening[31] | Achieving Universal Health Coverage (UHC) in Samoa through Revitalizing Primary Health Care (PHC) and Reinvigorating the Role of Village Women Groups<br>Baghirov et al[32] |
| Module 3: Community Engagement in All-Hazards Emergency and Disaster Risk Management | Sendai framework for disaster risk reduction 2015–2030[33]<br>Health Emergency and Disaster Risk Management Framework[34] | Shifting Paradigms: Strengthening Institutions for Community-Based Disaster Risk Reduction and Management<br>Bawagan et al[35] |
| Module 4: Community Engagement as a Driver for Achieving Health Equity and Community Resilience | Minimum Quality Standards and Indicators for Community Engagement[36] | Integrated vector control of Chagas disease in Guatemala: a case of social innovation in health<br>Castro-Arroyave et al[37] |

intensifying collaboration. The use of online platforms enabled the team to engage and mobilise relevant resources and develop the CEP components despite the absence of face-to-face consultations and other limitations. Creative use of online platforms was also maximised for the different components of the CEP (eg, online database, online modules) while still providing templates for possible face-to-face delivery, allowing for flexibility in engagement methods.

The three components of the CEP feed into each other. The thematic analysis of the materials in the CE database guided the design of the CELP and CEWP. Selected cases were also used to reinforce and provide real-world application to the CE frameworks and related concepts in the online modules. The CEWP facilitates the discussion of CE principles and practices among practitioners and the collection of new information for updating the database and CELP with 'new' CE experiences.

The merit of the current CEP project over existing documentation is that the CEP is broad based—not limited to health emergencies, but includes other public health and social developmental activities such as routine immunisation, neglected tropical diseases, city and urban development, nutritional interventions and disaster risk management, among others.

An operational challenge during the documentation was the language barrier. The cases were limited to English, French and Spanish. Future researchers can explore relevant documented and undocumented experiences in other languages, which will make the database more comprehensive and unifying at the same time.

## CONCLUSION

The design of the CEP emphasised inter-relationships among its components—CE database, learning package and workshop package. The CELP contents were taken from the comprehensive thematic analysis of the database. The CEWP facilitates the documentation of 'new' CE experiences and their nuances, ensuring timely updates of the database by CE practitioners themselves.

Most of the cases included in the CEP database presented key insights on CE including its basic principles and the role of individual and community agency, building on local capacity, multidirectional communication, inclusion and multistakeholder engagement. Barriers to CE including issues of access, acceptability and adoption in the setting of weak health systems and societal issues were also identified. The learning and workshop packages were then developed to guide health professionals and other stakeholders based on these grounds.

The development of the CEP was the work of multiple global stakeholders providing synergistic contributions and bridging silos. The description of the CEP methodology will allow replication, provide transparency into the development of the CEP and present lessons learnt during the development of a robust and harmonised package.

**Author affiliations**
[1]Country Readiness Strengthening, WHO, Lyon, France
[2]School of Health Sciences, University of the Philippines Manila, Manila, Philippines
[3]University of the Philippines Manila, Manila, Philippines
[4]Field Presence, WHO, Abuja, Nigeria

[5]Institute of Health Policy and Development Studies - National Institutes of Health, University of the Philippines Manila, Manila, Philippines

[6]Alliance for Improving Health Outcomes, Quezon City, Philippines

[7]Centro Internacional de Entrenamiento e Investigaciones Médicas (CIDEIM), Cali, Colombia

[8]Universidad Icesi, Cali, Colombia

[9]Hospital Escuela, Tegucigalpa, Honduras

[10]Instituto de Enfermedades Infecciosas y Parasitologia Antonio Vidal, Tegucigalpa, Honduras

[11]College of Medicine, University of Malawi, Blantyre, Malawi

[12]Kamuzu University of Health Sciences, Blantyre, Malawi

[13]Department of Clinical Pharmacy and Pharmacy Management, Nnamdi Azikiwe University, Awka, Nigeria

[14]Social Innovation in Health Initiative (SIHI), Nnamdi Azikiwe University, Awka, Nigeria

[15]Department of Parasitology and Entomology, Nnamdi Azikiwe University, Awka, Nigeria

[16]SIHI Honduras Hub, Tatumbla, Honduras

[17]Centro de educación medica continua Honduras, Tegucigalpa, Honduras

[18]Universidad Nacional Autónoma de Honduras (UNAH), Tegucigalpa, Honduras

[19]Facultad de Ciencias Médicas, Universidad Católica de Honduras Nuestra Señora Reina de la Paz Facultad de Ciencias de la Salud, Tegucigalpa, Honduras

[20]Cirugía General, Hospital General Santa Teresa, Comayagua, Honduras

[21]Evangelical Lutheran Development Services, Lilongwe, Malawi

[22]Maternal and Newborn Care, Joyful Motherhood, Lilongwe, Malawi

[23]Department of Clinical Epidemiology, University of the Philippines Manila, Manila, Philippines

**Acknowledgements** The authors would like to thank Dr Nedret Emiroglu, Director, WHO Country Readiness Strengthening Department, WHO; the WHO CEP Working Group members and resource persons: Qudsia Huda, Ankur Rakesh, Sohel Saikat, Saqif Mustafa, Samar Elfeky, Mervat Gawrgyous, Suvajee Good, Alex Camacho, Julienne Ngoundoung Anoko, Aphaluck Bhatiasevi, Faten Ben Abdelaziz, Ana Gerlin Hernandez Bonilla, Mihai Mihut, Nicole Valentine, Mary Manandhar, Dayo Spencer-Walters, Philippe Eric Gasquet, Joao Jose Salavessa Rangel De Almeida, Nina Gobat, Redda Seifeldin, Sameera Suri, Renee Christensen, Melinda Frost, Simon Van Woerden, Cristiana Salvi, Leonardo Palumbo, Aminata Grace Kobie, Anna Coates, Sonja Caffe, Gerry Eijkemans, Orielle Solar Hormazabal, Tonia Rifaey, Peggy Edmond Hanna, Godfrey Yikii, Dalia Samhouri, Supriya Bezbaruah, Kira Fortune and UNICEF resource persons Rania Elessawi, Naureen Naqvi, Ana Puri; the RCCE Collectives Services colleagues from the International Federation of the Red Cross and Red Crescent Societies (IFRC), UNICEF, Global Outbreak Alert and Response Network (GOARN) and WHO; and the CEP project staff, consultants and research associates: Erlyn Sana, Nina Castillo-Carandang, Gladys Armada, Jennel Pimentel, Justin Bryan Maranan, Nathalia Palma, Philippe Galban, Clarence Diaz, Eric David Ornos, Celina Gonzales, Samuel Tristan Vinluan, Ma. Pamela Tagle, Linda Mipando-Nyondo, Deborah Nyirenda, Katusha de Villiers, José Alejandro Carias, Perla Simons Morales, Sandra Barahona, Karla Zúniga, and Milena Bautista.

**Contributors** YVB, ML, JRBC, JDM-A, UEO, AOJ and NJ conceptualised and designed this work. JRBC, JDM-A, PMPT, AU, MSF, MIE, JA, DM, BKM, OIE, ON, LAL, NGQ, CINA, BYB, EDCR, ESN, VN-K and GM-K gathered and analysed data. YVB, UEO, SAO, JRBC, JDM-A and PMPT drafted the manuscript. YVB and ML are responsible for the overall content as guarantors. All authors reviewed, edited and approved the final version of the manuscript.

**Funding** The development of the WHO Community Engagement Package was funded by the WHO. The Social Innovation in Health Initiative (SIHI) is funded by TDR, the Special Programme for Research and Training in Tropical Diseases co-sponsored by UNICEF, UNDP, the World Bank and WHO. TDR receives additional funding from the Swedish International Development Cooperation Agency (Sida), to support SIHI (Grant/Award Number: N/A).

**Disclaimer** The authors alone are responsible for the views expressed in this article, and they do not necessarily represent the decisions or policies of PAHO or TDR. In any reproduction of this article there should not be any suggestion that PAHO or TDR endorse any specific organisation services or products.

**Competing interests** None declared.

**Patient and public involvement** Patients and/or the public were involved in the design, or conduct, or reporting, or dissemination plans of this research. Refer to the Methods section for further details.

**Patient consent for publication** Not required.

**Ethics approval** The development of the CEP did not entail participation of human subjects that requires ethical approval by the WHO Ethics Review Committee. The collection of feedback from pilot participants is a regular mechanism to evaluate training. Informed consent was obtained before documenting CE practitioners' experiences and recording workshop proceedings. Information about the project and its objectives and the extent of their participation was discussed. Regular internal SIHI and WHO reviews and consultative processes were facilitated to ensure that project deliverables met the needs of the end users and fulfilled the objectives of the project.

**Provenance and peer review** Not commissioned; externally peer reviewed.

**Data availability statement** Data are available upon reasonable request.

**ORCID iDs**
Jose Rene Bagani Cruz http://orcid.org/0000-0001-7539-0144
Jana Deborah Mier-Alpaño http://orcid.org/0000-0003-4119-8300
Ukam Ebe Oyene http://orcid.org/0000-0003-2632-7370
Allan Ulitin http://orcid.org/0000-0003-4190-501X
Marvinson See Fajardo http://orcid.org/0000-0003-3096-2801
Maria Isabel Echavarria http://orcid.org/0000-0002-9007-3507
Natalia Gomez Quenguan http://orcid.org/0000-0001-6858-328X
Claudia Ivette Nieto Anderson http://orcid.org/0000-0002-5301-2609
Eduardo Salomón Núñez http://orcid.org/0000-0003-1254-6724

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
