## [Reviewer comments · BMJ Open]

This paper was submitted to a another journal from BMJ Innovations and was then transferred to the BMJ Open for publication following peer review. The authors addressed the reviewers' comments and submitted the revised paper to BMJ Open. The paper was subsequently accepted for publication at BMJ Open.

(This paper received one review from its previous journal and one reviewer agreed to published their review.)

ARTICLE DETAILS

TITLE (PROVISIONAL)	A Description of Global Innovative Methods in Developing the WHO Community Engagement Package
AUTHORS	Bayugo, Yolanda; Labarda, Meredith; Cruz, Jose Rene Bagani; Mier-Alpano, Jana Deborah; Tiangco, Pauline Marie; Oyene, Ukam; Omoleke, Semeeh; Ulitin, Allan; Ong Jr., Alberto; Fajardo, Marvinson; Echavarria, Maria Isabel; Alger, Jackeline; Mathanga, Don; Msiska, Barwani; Ekwunife, Obinna Ikechukwu; Nwaorgu, Obioma; Abella Lizcano, Lorena; gomez quenguan, Natalia; Nieto Anderson, Claudia; Beltran, Briana; Carcamo Rodriguez, Elsy; Núñez, Eduardo; Nkosi-Kholimeliwa, Vera; Mwafulirwa-Kabaghe, Glory; Juban, Noel

VERSION 1 – REVIEW

REVIEWER	David, Annette Health Partners, LLC
REVIEW RETURNED	25-Jan-2022
GENERAL COMMENTS	Reviewer: 1 Comments to the Author This manuscript documents the process of creating learning tools for training on community engagement. The developmental process is outlined clearly and logically, and the modules developed comprising the CE learning package are delineated. The manuscript would benefit from a description of key evaluation findings from the "Participant feedback on the practical usefulness of the training contents, as well as their satisfaction with the methodology implemented by the course", which the authors note "were obtained through online evaluation forms."

VERSION 1 – AUTHOR RESPONSE

Reviewer: 1

This manuscript documents the process of creating learning tools for training on community engagement. The developmental process is outlined clearly and logically, and the modules developed comprising the CE learning package are delineated. The manuscript would benefit from a description of key evaluation findings from the "Participant feedback on the practical usefulness of the training contents, as well as their satisfaction with the methodology implemented by the course", which the authors note "were obtained through online evaluation forms."

Thank you for this recommendation. We have added the following paragraphs to the Results section to present the key findings from the feedback of the pilot participants.

Under the Community Engagement Learning Package:

The pilot participants found the CELP to be comprehensive in terms of content and with a user-friendly format. They appreciated how other concepts in public health were linked to CE. They suggested more practical applications and specific how-to's, and assessment activities with immediate feedback. These were all taken into consideration in the revision of the modules.

Under the Community Engagement Workshop Package:

Participants and observers of the CEWP pilot were satisfied with the introduction and ice-breaking activities which set the stage for conducive training sessions. Participants also expressed satisfaction on the content, pointing out that the workshop addressed aspects of CE not previously considered. The topics of the training were noted to be far-reaching, covering several CE frameworks, with good video presentations. Participants were able to relate the lessons and case studies to their experiences. They pointed out a few areas of improvement, including the need for adequate time to study the cases prior to the synchronous online sessions and more breakout sessions for participants to raise issues and ensure more diverse voices and opinions. They also recommended that the frameworks need to further emphasize listening and understanding community perspectives right from the start of the engagement.